# Peer-to-Peer Federated Learning for COVID-19 Detection Using Transformers

Mohamed Chetoui and Moulay A. Akhloufi *

Perception, Robotics, and Intelligent Machines Research Group (PRIME), Department of Computer Science, Université de Moncton, Moncton, NB E1A 3E9, Canada; emc7409@umoncton.ca
* Correspondence: moulay.akhloufi@umoncton.ca

**Abstract:** The simultaneous advances in deep learning and the Internet of Things (IoT) have benefited distributed deep learning paradigms. Federated learning is one of the most promising frameworks, where a server works with local learners to train a global model. The intrinsic heterogeneity of IoT devices, or non-independent and identically distributed (Non-I.I.D.) data, combined with the unstable communication network environment, causes a bottleneck that slows convergence and degrades learning efficiency. Additionally, the majority of weight averaging-based model aggregation approaches raise questions about learning fairness. In this paper, we propose a peer-to-peer federated learning (P2PFL) framework based on Vision Transformers (ViT) models to help solve some of the above issues and classify COVID-19 vs. normal cases on Chest-X-Ray (CXR) images. Particularly, clients jointly iterate and aggregate the models in order to build a robust model. The experimental results demonstrate that the proposed approach is capable of significantly improving the performance of the model with an Area Under Curve (AUC) of 0.92 and 0.99 for hospital-1 and hospital-2, respectively.

**Keywords:** federated learning; Vision Transformers; deep learning; COVID-19; medical imaging

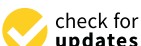



## 1. Introduction

Deep neural networks (DNN) consist of many layers with billions of parameters and are trained to learn complex, high-dimensional mappings from raw input data to output labels [1]. A major problem in training deep neural networks in the medical field is the huge amount of heterogeneous data required. A neural network trained on a single set of data from a single client can easily become overfitted, leading to bias towards that client and poor generalization. Additionally, potential patterns in the client image data can affect neural network performance in ways unrelated to the actual biological pathways in the image. The quality of data depends on many factors, including the number of patients, type or number of imaging equipment available, and number of specialists available at that institution (client). Deep learning models are usually trained on limited datasets because not all medical institutions have a large amount of diverse image data. This complicates clinical decision-making given the small number of cases that are more common in rare diseases. Deep learning models use large amounts of training data to make useful predictions. However, companies in highly regulated industries are reluctant to risk using or sharing sensitive data to create DNN models that promise uncertain rewards.

In healthcare, privacy laws and a fragmented market are preventing the industry from realizing the full potential of artificial intelligence (AI). Federated learning (FL) allows companies to train distributed models together without sharing sensitive medical records. From lung scans to brain MRIs, the large-scale aggregation and analysis of medical data could lead to new ways to detect and treat cancer, among other things.

Federated Learning is used to train distributed models across multiple devices, from smartphones to wearable medical devices, vehicles, and IoT devices. They help

build a robust model, but the training data is stored locally rather than shared, thus solving important issues such as data protection, data security and data access rights.

## 2. Related Work

Deep learning approaches are used on chest X-ray (CXR) images to classify patients by the presence or absence of COVID-19 infection, and the results have been demonstrated to be very good in accuracy (ACC), Area Under Curve (AUC), sensitivity (SN) and specificity (SP) terms. Ayalew et al. [2] presented a hybrid approach combined with a convolutional neural network (CNN) and histogram of oriented gradients (HOG), called DCCNet for a COVID-19 diagnosis using CXR images. Their hybrid model achieved an accuracy score of 99.67%. Ghose et al. [3] presented the transfer learning for COVID-19 detection using CT-scan and CXR images. The authors merged CT-scan with CXR images to create a global dataset. Their algorithm obtained an accuracy score of 99.59% for CXR and 99.95% for CT-scan images. Indumathi et al. [4] presented a method based on a machine learning (ML) algorithm to identify the degree of infection of COVID-19; the ML algorithm classified COVID-19-affected regions into various zones such as: danger, moderate, and safe zones. Their proposed approach obtained an accuracy score of 98.06%. Salau et al. [5] provided a Support Vector Machine (SVM) algorithm for the identification and classification of COVID-19. The authors used a discrete wavelet transform (DWT) algorithm for feature extraction and SVM for classification. Their method achieved an accuracy score of 98.2%. Frimpong et al. [6] presented an interesting study for COVID-19 detection based on a Wi-Fi-enabled microcontroller, temperature sensor, and heart rate sensor. The authors designed a low-cost hardware system for students. The suggested method monitored the student's condition continuously on a mobile application while detecting and differentiating between a normal and abnormal body temperature and a regular and irregular heartbeat. Tests over time demonstrated the created IoT-enabled system's dependability, responsiveness, and affordability. The microcontroller's intelligent programming and the sensor's operation is through the mobile application, which enables the low-cost early diagnosis of abnormal temperature and heartbeat anomalies.

Lua et al. [7] presented a multiscale class residual attention (MCRA) network for the multiclass classification of COVID-19, pneumonia, and normal cases using CXR images. The authors used pixel-level image mixing of local regions for data augmentation and reducing the noise. Their experimental results demonstrated that their network achieved an accuracy score of 97.71%.

Chouat et al. [8] presented a series of pre-trained DL models: ResNet50, InceptionV3, VGGNet-19, and Xception for COVID-19 detection on CXR and CT-scan images. The authors included a data augmentation technique to increase the size of the dataset. Authors found that the VGGNet-19 outperformed the other three DL models on the CT image dataset, where it achieved an accuracy score of 87.0%, and their best model for CXR images was the Xception, with an accuracy score of 98.0%.

Abdul et al. [9] studied the efficacy of FL versus traditional learning by developing two machine learning models; the authors used descriptive dataset CXR images from COVID-19 patients. They tried to identify which factors affect model prediction loss and accuracy (ACC) by changing the hyperparameters. Their results demonstrated that the FL model has a better score in terms of ACC and loss compared to the traditional machine learning model.

Boyi et al. [10] proposed an experiment to compare the performance of federated machine learning, using four CNNs models (MobileNet, MobileNet-v2, ResNet18, and COVID-Net), by training them in a CXR dataset. The models are developed to classify COVID-19 disease; after 100 rounds, the authors demonstrated that the ResNet18 model gives the highest ACC score of 96.15%.

Junjie et al. [11] proposed an FL system based on the digital twin city concept to study the impact of different city prevention plans to contain the COVID-19 outbreak. They developed an FL model to predict the effect. They were able to track the infection number

from various cities over time using their digital city twin systems. Moreover, the authors were able to track the success of every preventative strategy by creating a regional model on every digital city and sending the model's to a federated server to protect data privacy. Finally, they developed a comparison between the traditional model and the federated one using mean squared error (MSE) loss; the minimum loss was 0.01 for federated learning and 0.2 for non-federated learning.

To improve the performance measures for FL models, Weishan et al. [12] introduced a novel dynamic fusion-based FL approach for COVID-19 detection. They found that recent FL studies employed the default FL settings, which might result in significant communication overhead and perform poorly when clients' data are heterogeneous. Authors suggested a method that uses a dynamic fusion-based function to identify the interaction between clients and servers to decide which client takes part in each round of uploading his local model updates. The platform owner specified a maximum waiting period for each client to take part in the server round. The results demonstrated that for ResNet50 and ResNet101, their approach introduced better ACC than the default settings, with a score of 96%.

A system for real-time COVID-19 case detection and monitoring was suggested by Mwaffaq et al. [13]. Five machine learning methods were shown to have higher than 90% prediction ACC in their study, which used an IoT device for data collecting and monitoring during quarantine. They used seven machine learning algorithms, ran experiments on each algorithm, and compared them.

Dayan et al. [14] used data from 20 institutes across the globe to train the FL model, called EXAM (electronic medical record (EMR) chest X-ray AI model). For COVID-19 detection, they used inputs of vital signs, laboratory data, and CXR images. They achieved an AUC of 0.92 for predicting COVID-19.

In this study, we provide an approach for FL in the medical field named peer-to-peer federated learning (P2PFL), which allows clients to share their model without the intermediary of a central server, in order to build a robust model and avoid problems caused by the server. We train the models with Vision Transformers (ViT) networks in order to classify COVID-19 and normal patients using two CXR datasets. Each dataset is assumed to be assigned to a hospital. Our study solves the problem of confidentiality and data security. Finally, we compare the performance of the models trained with P2PFL and non-federated learning. Figure 1 gives an overview of our methodology.

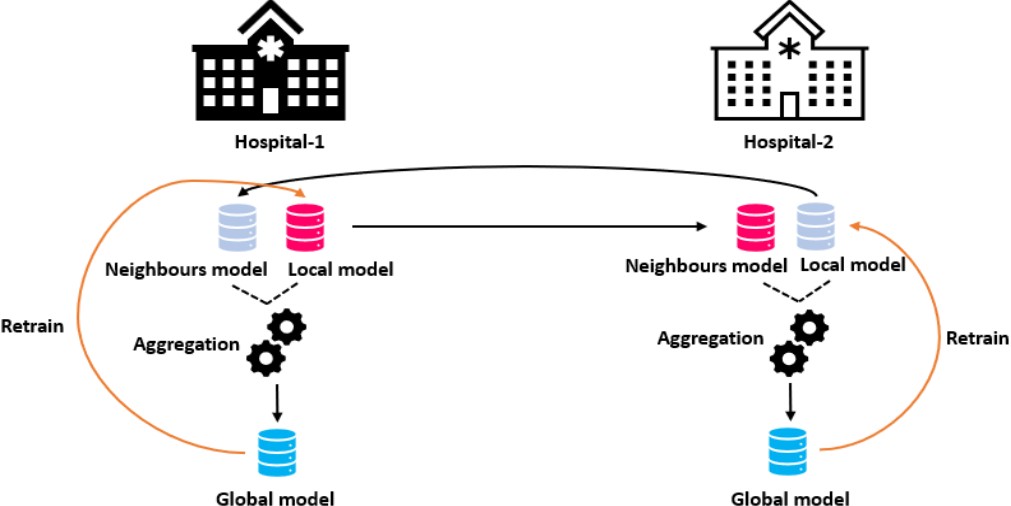

**Figure 1.** Proposed peer-to-peer federated learning learning architecture for COVID-19 detection.

## 3. Federated Learning

Deep learning (DL) models are usually trained centrally, with data stored at the client site, and model owners are free to observe the data. Collecting diverse and rich datasets can

be challenging in many cases due to data sensitivity. This makes it difficult to train robust deep learning models that require sufficiently large and diverse datasets [15]. To resolve this problem, McMahan et al. [16] present FL which decentralizes the training of machine learning models. In FL, clients can participate in the training of an DL model; each client trains a model using a local dataset, then they share the model parameters with other clients. The authors use a weighted average of the models, a technique called 'Fedavg'.

In Fedavg, the global model is first initialized. Then in every round $t$, the central server sends the current global model $wt$ to a selected fraction $C$ of all clients $K$. The selected clients are expressed as the set $St$. Each client $k$ then trains the model on its local data $Pk$, resulting in a model $w_{t+1}^k$, and sends its updated local model parameters to the server. Then, the server aggregates the received models' weights to generate a new global model using the following equation:

$$w_{t+1} = \sum_{k \epsilon S_t} \frac{n_k}{n_t} w_{t+1}^k$$

where $n_t$ represents the total number of all samples from the clients, and $n_k$ is the number of samples at client $k$. When the training is completed, the server sends the aggregated model to the clients in the network.

However, the presence of a server in the network can provide risks to clients. For example, the attacker might use a fine-tuning strategy to create some malicious updates that are then distributed from the central parameter server to compromise the entire local training group. Furthermore, federated learning frameworks that rely on a single parameter server pose a risk of failure. If the attacker compromises the main parameter server, training might be interrupted and/or stopped. To minimize the risk, we use P2PFL proposed by Lalita et al. [17]. There is no need for a central server; we adapt Fedavg to work in a peer-to-peer manner.

Each client has its model and communicates with connected neighbors. Before the training, all models of clients are initialized with the same weights $w_0$, and in every round $t$, each client $c$ trains the model on its local dataset $P_c$, resulting in a model $w_t^c$. Then, for each client in the network, updates are collected and averaged from a set of random neighbors $S_t$, where $|S_t|$ is determined by $C.A.$, and where $C$ is the proportion of neighbors and $A$ is the total number of neighbors in the network for that client. Then, the local model is updated as follows:

$$w_{t+1}^c = \frac{n_c \cdot w_t^k}{n_t} + \sum_{k \epsilon S_t} \frac{n_k}{n_t} w_t^k$$

where $n_k$ is the number of samples at neighbor $k$ and $n_t$ is the total number of samples from client $c$ and from the clients in $S_t$.

## 4. Client's Datasets

### 4.1. Hospital-1

The SIIM-COVID-19 dataset [18], which contains COVID-19 CXR images, is attributed to hospital-1. It is a public dataset made available by the Society for Imaging Informatics in Medicine. Each CXR image includes labeling for detecting COVID-19 anomalies on CXR. The 6334 CXR images in the dataset are in a DICOM format and have been de-identified to protect patient privacy. A group of expert radiologists classified each image based on its general appearance as well as the prevalence of opacities. The dataset also contains 1263 CXR images for the test.

We obtained the normal CXR images from RSNA dataset, which is provided by the US National Institutes of Health Clinical Center. It includes 26,684 CXR images for distinct individuals, and each image is categorized into one of three categories based on the radiology reports that are associated with it: "Normal", "No Lung Opacity/Not Normal", and "Lung Opacity".

We used 11,834 CXR images (5500 normal; 6334 COVID-19) for the model development and 2208 CXR (945 normal; 1263 COVID-19) for the test.

Figure 2 shows examples of SIIM-COVID-19 and RSNA CXR images.

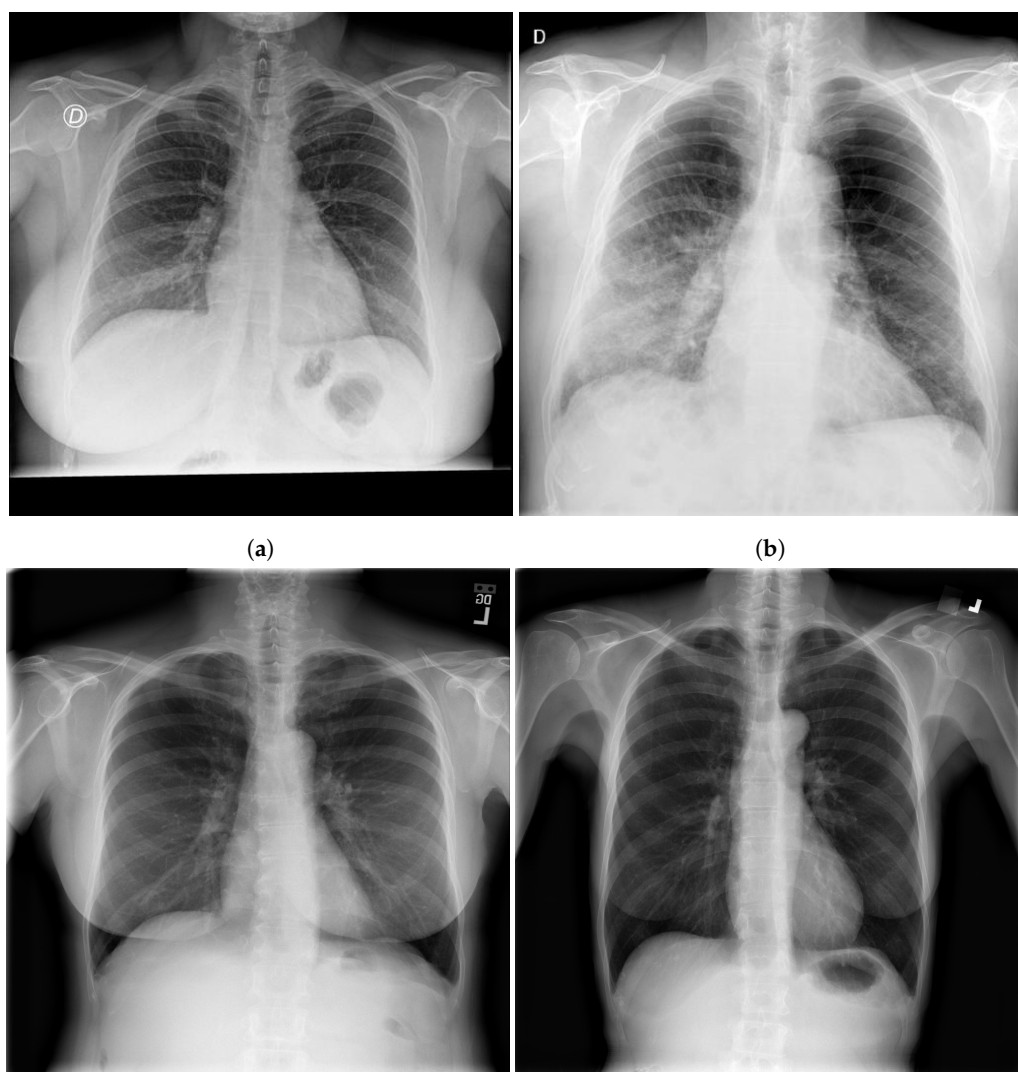

(a)

(b)

(c)

(d)

**Figure 2.** Examples of CXR images; SIIM-COVID-19: (**a**,**b**) and RSNA: (**c**,**d**).

*4.2. Hospital-2*

The BIMCV COVID+ dataset [19], which contains COVID-19 images, is attributed to hospital-2. It is a CXR dataset of patients with COVID-19, radiography findings, pathologies, PCR test results, immunoglobulin G (IgG) and immunoglobulin M (IgM) diagnostic antibody tests, and radiography reports from the Medical Imaging Databank in Valencia Region Medical Image Bank (BIMCV). The images are annotated and kept in high resolution by a group of qualified radiologists. Furthermore, a variety of data are offered, including, among other things, the patient's demographics, the projection type (PA-AP), and the imaging study acquisition parameters. A total of 1380 CXR, 885 DX (Digital X-ray), and 163 for the computed tomography are all present in the dataset.

The normal CXR images used for hospital-2 are imported from the NIH dataset [20]. It comprises 112,120 CXR images with disease labels from 30,805 unique patients. This dataset was obtained from the National Institute of Health (USA), and the expert physicians assigned grades to the CXR images.

We used 4145 CXR images (2200 normal; 1945 COVID-19) for the model development and 1036 CXR (553 for normal, 483 COVID-19) for the test.

Figure 3 shows examples of SIIM-COVID-19 and RSNA CXR images.

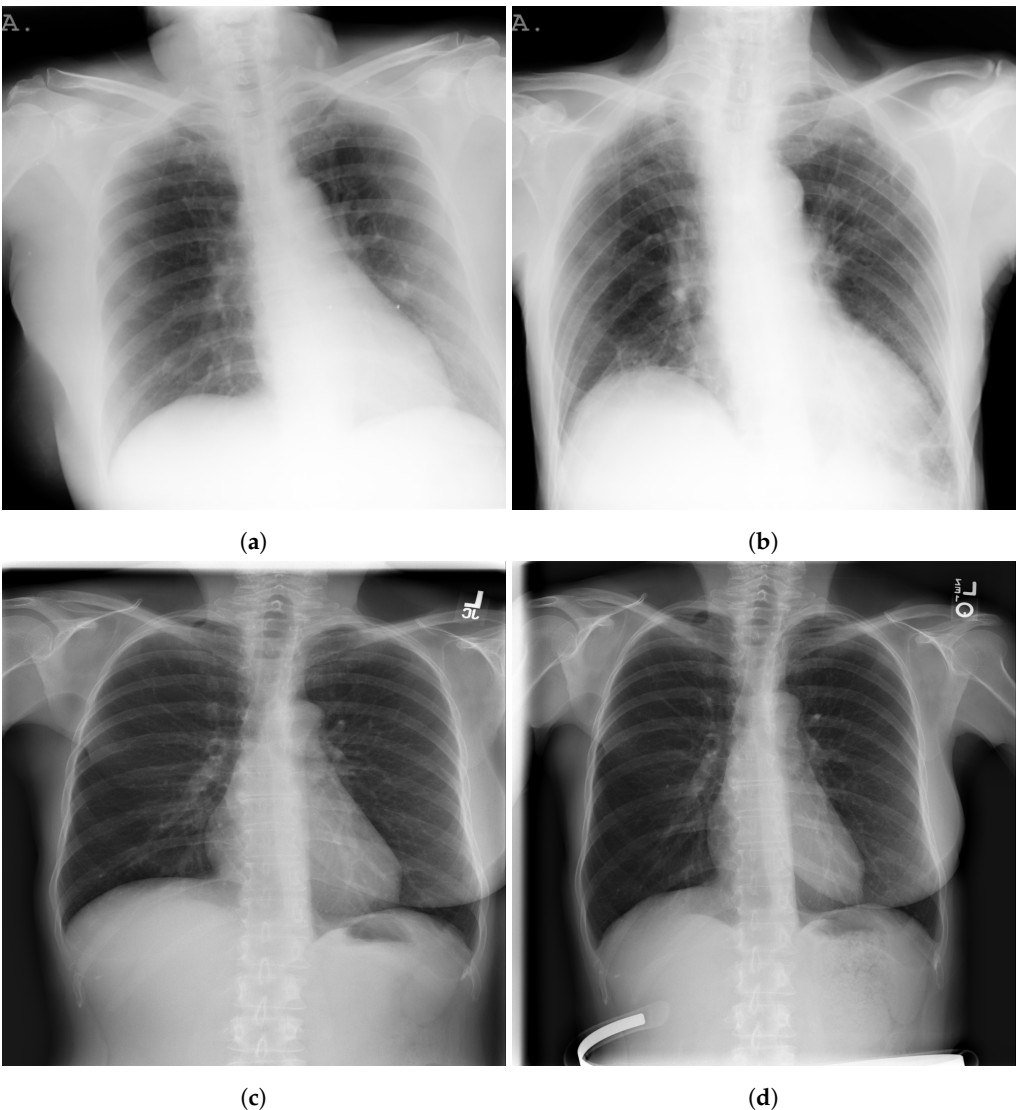

(a)

(b)

(c)

(d)

**Figure 3.** Examples of BIMCV COVID+; (**a**,**b**) and NIH; (**c**,**d**) CXR images.

## 5. Vision Transformer Models

A deep neural network based on an attention mechanism that makes use of a noticeably broad receptive field is called Vision Transformer (ViT). The capacity to model long-range dependency inside an image and achieve a state-of-the-art (SOTA) performance in NLP have encouraged the vision community to investigate its application on vision tasks [21]. When compared to SOTA convolutional neural networks in image classification tasks; the ViT was one of the successful attempts to apply Transformer directly on images [22]. ViT's simple modular design, in addition to its excellent performance, enables extensive applications in a variety of tasks with minimal modifications.

By dividing ViT into a shared body and task-specific heads and tails, Chen et al. [23] presented an image processing transformer, one of the successful multi-task models for diverse computer vision tasks. They used an encoder–decoder design. ViT was recently used to diagnose and predict the severity of COVID-19, demonstrating its SOTA performance [24].

There are several variants of Transformers; in this study, we use the ViT-B32 model in each hospital. The reason for choosing this architecture is because it gives very interesting results for the detection of COVID-19 in CXR images [25].

The classification head for ViT is represented by an MLP during pre-training and by a classification portion during fine-tuning. We added a flatten layer, a batch-normalization

layer, a dense layer of size 11, and then another layer of batch-normalization to each ViT model. The Softmax function provides the probability of classifying the CXR images as normal or COVID-19. Non-P2PFL was locally trained using the same model of P2PFL hyperparameters and 100 epochs on the same data division.

## 6. Metrics

For performance evaluations, we used the following metrics: Accuracy (*ACC*), Sensitivity (*SN*), and Specificity (*SP*). These measures are described as follows:

$$SN = \frac{TP}{TP + FN} \tag{1}$$

$$SP = \frac{TN}{TN + FP} \tag{2}$$

$$ACC = \frac{TP + TN}{TP + FN + TN + FP} \tag{3}$$

where $TP$ is the True Positive rate and means the number of positive cases that are correctly labeled, $TN$ is for the True Negative rate and represents the number of negative cases that are correctly labeled, $FP$ is for the False Positive rate and represents the number of positive cases that are falsely labeled, and $FN$ is for the False Negative rate. Area Under Curve (AUC), which is calculated using the ROC curve, is a performance metric frequently used for medical classification issues to demonstrate where the model compromises between accurate and unreliable classifications.

## 7. Results

In this section, we present the results obtained in each hospital for COVID-19 detection using the P2PFL technique and Non-P2PFL.

Keras Library [26] was used to develop the P2PFL and ViT models. The training was carried out using Nvidia P6000 [27]. RectifiedAdam [28] was used as an optimizer, and the Batch size was fixed to 32 for each model with 100 rounds. All CXR images were resized to 224 × 224.

Figure 4 presents the learning curves (train and validation) for hospital-1 and 2 using 100 rounds. As we can observe, for hospital-1, in every end of 20 rounds, the model achieved a high ACC score and tried to learn more important features. In the final round, the model obtained an ACC of 0.85 using the P2PFL method and 0.83 for Non-P2PFL, which is an improvement by 2%. It was the same for hospital-2; after 100 rounds, the model obtained an interesting ACC score of 0.98 using the P2PFL technique compared to Non-P2PFL, which gives an ACC score of 0.97. The confusion matrix for P2PFL is shown in Figure 5b and only 134 CXR images of COVID-19 were misclassified; the Non-P2PFL technique demonstrates 187 of FN, which is a higher number compared to P2PFL (*FN* = 134) (see Figure 5a). This shows that the P2PFL technique helps the model to improve its performance. The confusion matrix for hospital-2 using P2PFL is shown in Figure 6b. As demonstrated, the P2PFL technique improves the performance of the model, and only 21 images were misclassified compared to Non-P2PFL (24 of misclassified images). Table 1 summarizes the performance scores obtained for each hospital for P2PFL and Non-P2PFL techniques. Figures 7 and 8 shows the AUC curves of hospital-1 and hospital-2. For hospital-1 (Figure 7) an AUC score of 0.92 for P2PFL technique and 0.89 for Non-P2PFL, (less than P2PF by 3%). For hospital-2 (Figure 8), the model obtained an AUC score of 0.99 for P2PFL and 0.98 for Non-P2PFL, this shows that the suggested technique is a significant improvement.

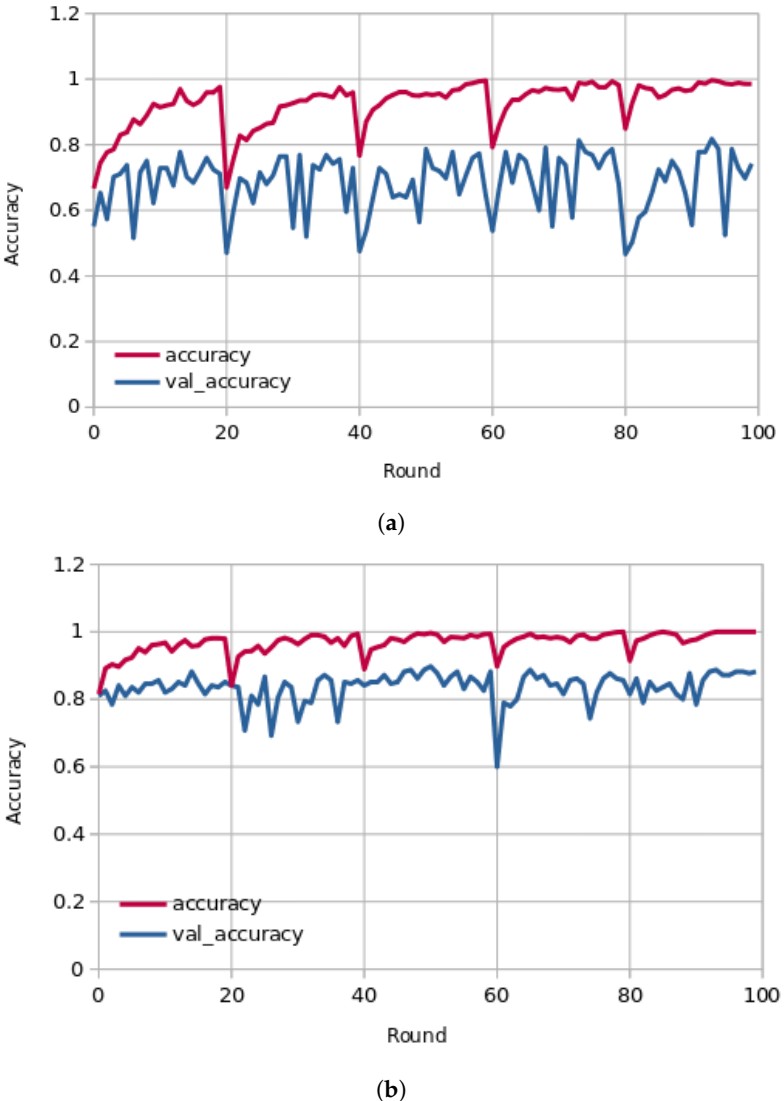

(**a**)

(**b**)

**Figure 4.** Train and validation curves of P2PFL for hospital-1 and 2. (**a**) hospital-1; (**b**) hospital-2.

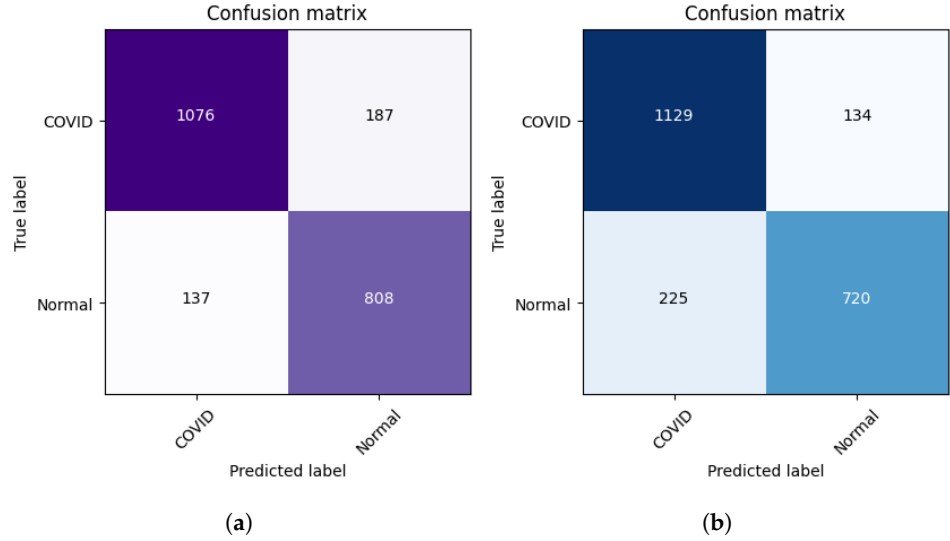

(**a**)                                                                                              (**b**)

**Figure 5.** Confusion matrices of hospital-1 Non-P2PFL (**a**); P2PFL (**b**) (COVID-19 vs. normal) classification.

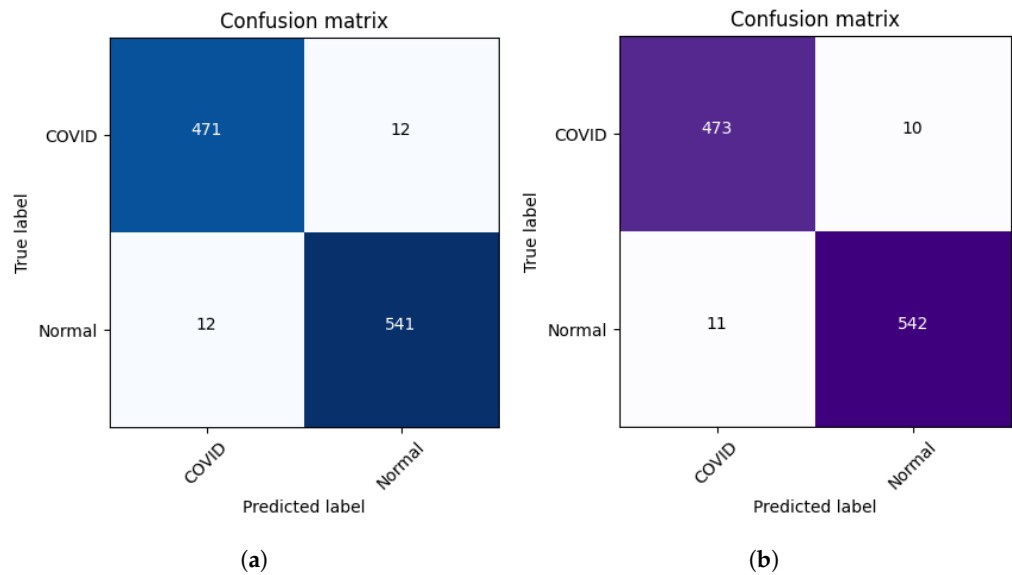

**Figure 6.** Confusion matrices of hospital-2 Non-P2PFL (**a**); P2PFL (**b**) (COVID-19 vs. normal) classification.

**Table 1.** Performance measures using P2PFL vs. Non-P2PFL with Vision Transformer models.

| Clients | P2PFL | | | | Non-P2PFL | | | |
|---|---|---|---|---|---|---|---|---|
| | ACC | AUC | SP | SN | ACC | AUC | SP | SN |
| **Hospital-1** | 0.85 | 0.92 | 0.85 | 0.85 | 0.83 | 0.89 | 0.84 | 0.84 |
| **Hospital-2** | 0.98 | 0.99 | 0.98 | 0.98 | 0.97 | 0.98 | 0.97 | 0.97 |

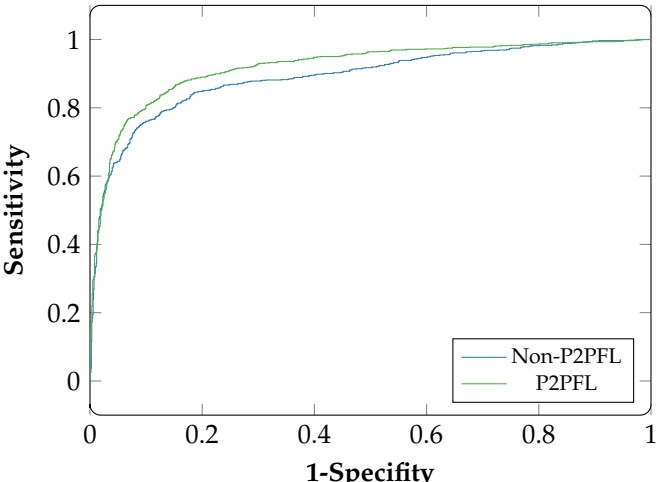

**Figure 7.** ROC curves of Non-P2PFL and P2PFL for hospital-1 classification (COVID-19 vs. normal).

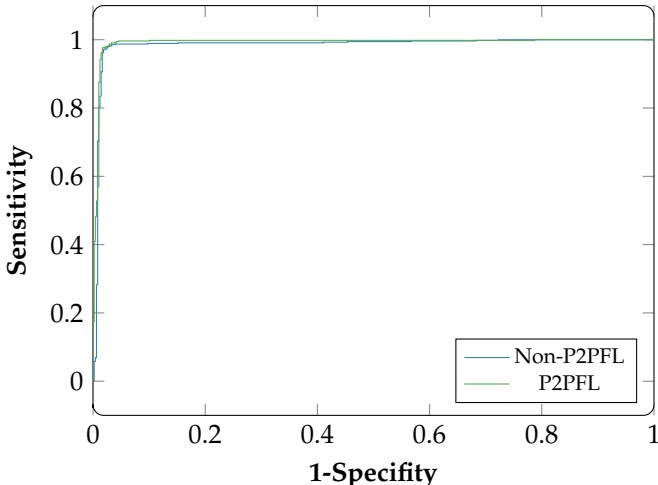

**Figure 8.** ROC curves of Non-P2PFL and P2PFL for hospital-2 classification (COVID-19 vs. normal).

## 8. Model Explainability

A Self-Attention with MLP module and a Standard Transformer Encoder make up the visual transformer. We visualised the signs recognized by ViT using the attention map of ViT-B32 in order to better understand how the model learned to recognize the COVID-19 signs. The self-attention score for the model can be used to visualize the input image. In this investigation, the jet colour scheme is employed. The blue tones in this colour scheme indicate lower values, which indicate that no features are extracted for a particular class, while the yellow and green tones indicate medium values, which demonstrate an intermediate probability for being in a particular class; the red and dark red tones indicate higher values, which indicate that the features in the region correspond to the particular class. Figure 9a shows samples of true positives with the attention map. The images demonstrate that the affected area was highlighted more precisely by the P2PFL than by non-P2PFL model. This shows that after the model aggregation of the two hospitals, we obtain a robust model to detect signs of COVID-19. The detection of signs becomes more accurate if high resolution CXR images are used.

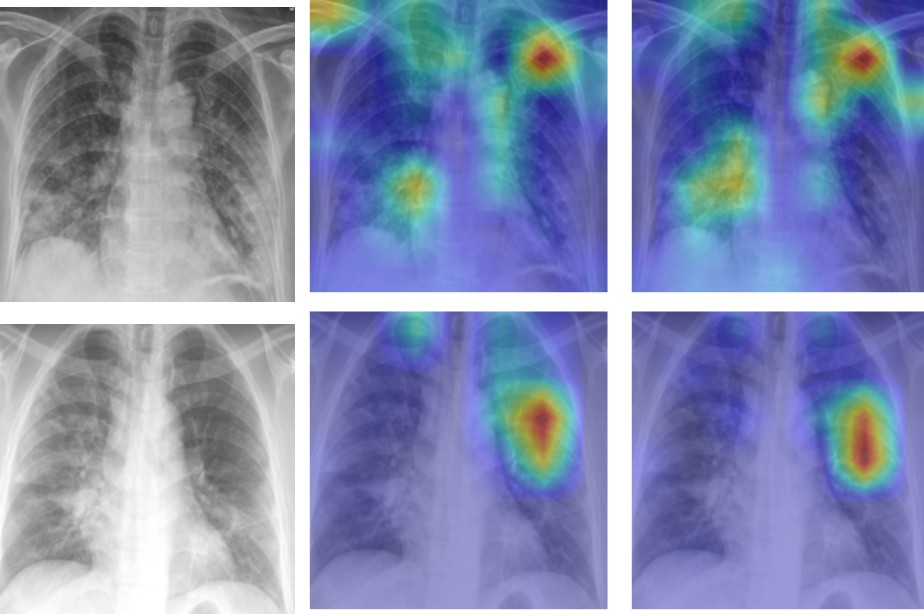

**Figure 9.** *Cont.*

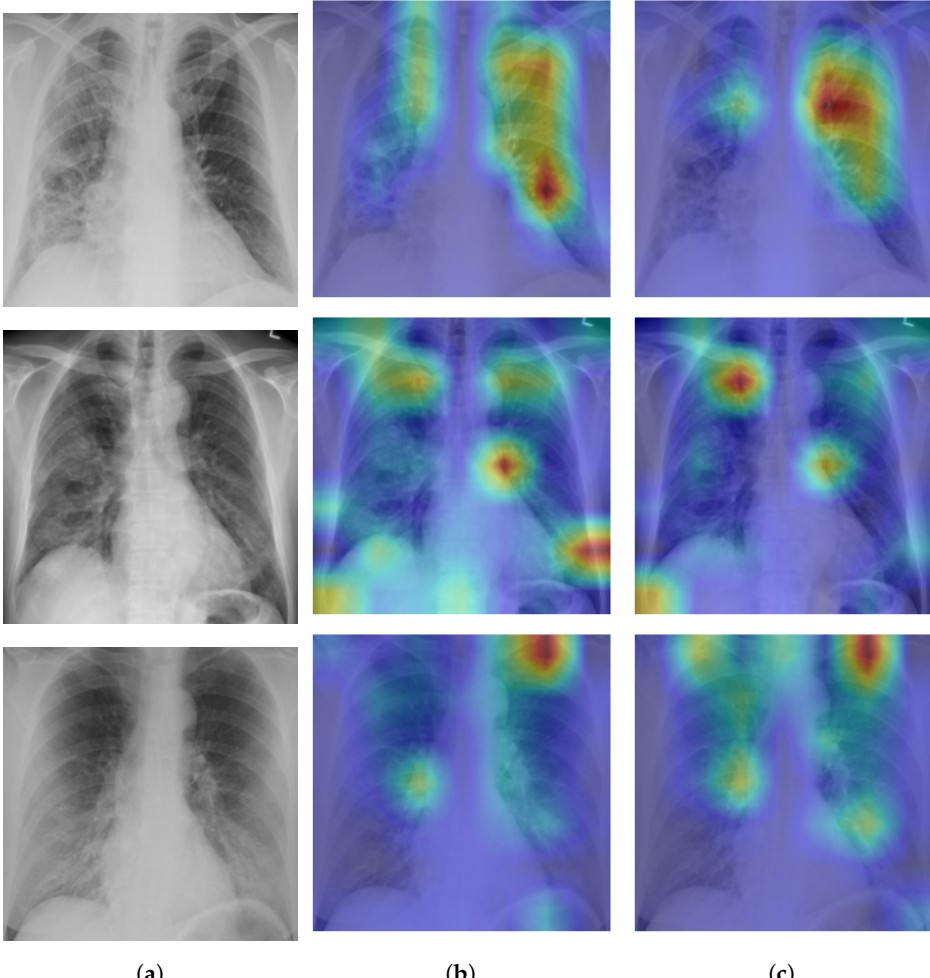

<div align="center">(<b>a</b>)       (<b>b</b>)       (<b>c</b>)</div>

**Figure 9.** Example of explainability COVID-19 cases on hospital-1 and hospital-2 using P2PFL and non-P2PFL: column (**a**) original images, (**b**) TP COVID-19 cases for non-P2PFL, (**c**) TP COVID-19 for P2PFL.

## 9. Model Limitations

Although Transformers have been demonstrated to be good alternatives to CNNs, their application is still a bit challenging because of the requirement for huge datasets. In fact, CNNs can learn even when there is just a limited quantity of data available, largely due to the presence of inductive biases. This help speed up learning and improve generalization for models. In particular, CNNs contain two biases that are fundamental to how the architecture itself operates, namely:

- The adjacent pixels in the image are connected to one another.
- Regardless of their absolute positions, different components of the image must be processed in the same way.

Although they require more information to completely comprehend the issue, they are able to do it more easily because these biases are not present in the Transformer's architecture. Therefore, it may be claimed that whereas Convolutional Neural Networks attain a poorer comprehension of the issue at hand but also do it with smaller data sets, Transformers are able to learn more but require more input. Huawei et al. [29] presents an approach called Convolutional Neural Networks Meet Vision Transformers (CMT) Block. It combines CNN with Transformers. In addition to delivering some efficiency improvements, several of these blocks are used within a novel architecture that combines the self-attention and convolution mechanisms. Because of the different characteristics added in the various layers, the resulting design is able to effectively benefit from the best of both networks.

## 10. Discussion

The experimental results in Table 1 demonstrate that the proposed P2PFL approach achieves an exceeding performance in two hospitals in terms of ACC, AUC, specificity, and sensitivity. Aggregating the weights allows the model to be more robust in detecting even more characteristics, because the two clients use two completely different datasets in terms of image quality, brightness, orientation of the lungs, etc.

The techniques presented in this study offer several privacy benefits. In data sharing, the raw data remains on the local clients and only the updates of the model are exchanged between clients and connected clients. No non-aggregated data are kept on clients receiving aggregated models. Despite the benefits, there are several challenges in this field for further investigation. For example, in this study, we assume that all clients are honest and that there are no malicious clients. The impact of malicious clients propagating stealth backdoor attacks during training needs a new research direction to improve learning integrity.

This method offers a strong proof-of-concept that P2PFL may be used to efficiently and collaboratively develop the required AI models for the health sector. Several hospitals were included in our study. In comparison to any model trained with Non-P2PFL, the global P2PFL model was demonstrated to be more reliable and produced superior outcomes at individual hospitals. We believe that the more diversified dataset, the use of standard data inputs, and the exclusion of clinical impressions/reported symptoms were the key factors in the improvement. These features significantly increased the advantages of this P2PFL technique and its effects on performance, generalizability, and, ultimately, the usability of the model.

It is necessary to optimize clinical information systems in order to better use a network of P2PFL-affiliated sites by streamlining data preparation. This can help algorithms "learn" more efficiently from larger data batches and adapt model parameters to a specific site for additional customization. This is in addition to hyperparameter optimizing.

Knowing how much each client site will contribute to enhancing the global P2PFL model is a feature that would improve this kind of large-scale collaboration. This will help in choosing the client site and prioritizing the efforts to collect and annotate data.

The most effective training settings for each client site may be found more efficiently in the future using methods such as automated hyperparameter searching, neural architecture search, and other automated machine learning methodologies.

## 11. Conclusions

In this paper, we proposed peer-to-peer federated learning over a network of two clients (hospitals), with no server, using a Vision Transformers' model for COVID-19 detection. The clients jointly iterate and aggregate their models in order to build a robust model. Our study minimizes the risks that can be caused by the server and gives more protection to the model shared between the clients. This technique also improves the communication speed between the clients without an intermediary server. The architecture of the model was based on the Vision Transformer, demonstrated the performance of classification and detecting the important features of the disease. We tested the aggregated model on the test sets for each client. We obtained high AUC scores of 0.92 and 0.99 for hospital-1 and hospital-2, respectively. Our experiments demonstrate interesting results, and the proposed technique improved the performance of the model compared to non-peer-to-peer federated learning. Furthermore, if we can implement the suggested approach with a large site of clients, the efficiency increase may result in the increased performance of models. Future work may involve more clients to achieve better results, which would further improve the suggested technique. In addition, to increase the degree of data privacy, we aim to include extensive empirical studies with various techniques such as split federated learning.

**Author Contributions:** Conceptualization, M.C. and M.A.A.; methodology, M.C. and M.A.A.; software, M.C.; validation, M.C. and M.A.A.; formal analysis, M.C. and M.A.A.; investigation, M.A.A. and M.C.; resources, M.C.; data curation, M.C.; writing—original draft preparation, M.C. and M.A.A.; writing—review and editing, M.C. and M.A.A.; supervision, M.A.A.; project administration, M.A.A.; funding acquisition, M.A.A. All authors have read and agreed to the published version of the manuscript.

**Funding:** This work was partially supported by the Natural Sciences and Engineering Research Council of Canada (NSERC), Alliance Grants (ALLRP 552039-20), New Brunswick Innovation Foundation (NBIF) COVID-19 Research Fund (COV2020-042), and the Atlantic Canada Opportunities Agency (ACOA), Regional Economic Growth through Innovation-Business Scale-Up and Productivity (project 217148).

**Institutional Review Board Statement:** IRB université de Moncton waives the approval requirements for the data used in this work since they are from publicly available datasets.

**Informed Consent Statement:** Not applicable.

**Data Availability Statement:** The data used in this work come mainly from public datasets. Please see the section describing the datasets.

**Conflicts of Interest:** The authors declare no conflict of interest.

## Abbreviations

The following abbreviations are used in this manuscript:

| | |
|---|---|
| ACC | Accuracy |
| AI | Artificial intelligence |
| AUC | Area Under Curve |
| BIMCV | Medical Imaging Databank in Valencian Region Medical Image Bank |
| CXR | Chest-X-Ray |
| DL | Deep learning |
| DICOM | Digital Imaging and Communications in Medicine |
| DNN | Deep neural networks |
| DX | Digital X-ray |
| EMR | Electronic medical record |
| EXAM | Electronic medical record chest X-ray AI model |
| FL | Federated learning |
| FN | False negative |
| FP | False positive |
| IgG | Immunoglobulin G |
| IgM | Immunoglobulin M |
| IoT | Internet of Things |
| NIH | National Institutes of Health |
| Non-I.I.D. | Non-independent and identically distributed |
| Non-P2PFL | Non-peer-to-peer federated learning |
| P2PFL | Peer-to-peer federated learning |
| PCR | Polymerase chain reaction |
| RSNA | Radiological Society of North America |
| SN | Sensitivity |
| SP | Specificity |
| SIIM | Society for Imaging Informatics in Medicine |
| SOTA | State-of-the-art |
| TN | True negative |
| TP | True positive |
| ViT | Vision Transformer |

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
