# Peer review of "Peer-to-Peer Federated Learning for COVID-19 Detection Using Transformers"

_computers, doi:10.3390/computers12050106_

Round 1
Reviewer 1 Report
The paper proposes an interesting approach to tackl the problem of insufficient amount of data in medical problems. The authors propose the use of federated learning, so that data can be kept in its original source indstitution and still contribute to a deep learning process.
I have just a few questions.
1. The data available at different institution/hospital can have missing oroperties, how would you deal with it?
2. What is the computational cost of the additional computinb needed for accessing distributed data?
3 How would such an approach scale in terms of number of participating hospitals?
4. In this paper a vision transformer is trained, woulc it be transferable to other thoes of data? Other problems?
Author Response
Thank you for taking the time to review our paper, here is a point-by-point response to your comments and concerns.

Reviewer 2 Report
The paper "Peer-to-Peer Federated Learning for COVID-19 Detection using Transformers" proposes a novel approach to training deep neural networks on heterogeneous medical data using federated learning. The authors introduce a peer-to-peer federated learning (P2PFL) framework based on Vision Transformers (ViT) models for the classification of COVID-19 and normal cases on Chest-X-Ray images. The proposed framework addresses the challenges of data protection, security, and access rights that arise when working with sensitive medical data. The results of their experiments demonstrate that the P2PFL framework is efficient and effective in improving the model's performance, achieving an Area Under Curve (AUC) of 0.92 and 0.99 for hospital-1 and hospital-2, respectively.
However, I have the following concerns,
(1)To properly evaluate the performance of the proposed P2PFL framework, it would be helpful to have a clear understanding of how the Non-P2PFL model was trained. The authors have not provided this information, which creates uncertainty about the fairness of the comparison. It seems likely that the Non-P2PFL model was trained on data from each hospital individually, which would not provide the same opportunities for performance improvement as the P2PFL model. For instance, the P2PFL model can leverage training examples from other hospitals, which could be particularly beneficial for a data-hungry model like the Vision Transformers (ViT). In light of these concerns, I recommend that the authors provide more details about how the Non-P2PFL model was trained, or consider running additional experiments to ensure a fair comparison.
(2) I have some concerns about the toy setting used for P2PFL in this paper. With only two hospitals participating, it may not fully utilize the potential of P2PFL. For example, Reference 10 discusses P2PFL on graphs, where it makes sense for neighbors to learn from each other. In this case, it might be worth considering combining the data from both hospitals and training the Vision Transformers (ViT) on the combined dataset to see if better performance can be achieved.
(3) I am uncertain about the details of the ViT model used by the authors. While Chen et al. [16] use PyTorch in their implementation, it seems that the authors of this paper have used Keras. It would be beneficial if the authors could provide their code to allow for a better evaluation of their model. If the authors have developed their ViT model from scratch, I suggest that they consider open-sourcing it, as such a complex model requires a significant amount of effort to develop and optimize for practical use. This would include coding tricks, quality testing, training settings, and long-time training to obtain better weights.
(4) In lines 157 and 158, the author claims that the chosen architecture yields interesting results for COVID-19 detection in CXR images. However, no details are provided to support this claim. To enhance the reader's understanding, the authors should elaborate on why this architecture was selected and how it is particularly effective for detecting COVID-19 in CXR images. Providing more specific information could help readers better appreciate the significance of the authors' findings.
Some small errors,
(1) No space for the beginning of a new paragraph for Lines 117, and 136.
(2) In Line 95, I think it is a typo for `nt` and `nk`, it should be a subscript for `t` and `k`.
Author Response

(The authors gave the same response as above.)

Reviewer 3 Report
This paper presented a detailed approach for for COVID-19 detection using Transformers. Overall, the paper is well written. However, there are some suggestions for the final version:
1- The abstract needs revision. Too many details and generally lacks abstraction. It will be better to concentrate on the aim, contributions, methods, and what has been achieved.
2- The introduction section must clearly identify the problem, define why conducting this research is important and discuss what are the differences between this research and research stated in the literature.
3- The proposed model must be compared with the existing models published in recent years.
4- The author needs to proofread the paper.
5- In the conclusion section author must add the future direction of the research.
Author Response

(The authors gave the same response as above.)

Round 2
Reviewer 2 Report
Overall, the revision looks good. One minor thing to note is that there seems to be a typo at the beginning of line 200, where there is an extra '-'.
Author Response
Thank you very much for taking again the time to review our paper, the typo at the beginning of line 200 has been corrected